# The Effect of Older Age and Frailty on the Time to Diagnosis of Cancer: A Connected Bradford Electronic Health Records Study

**DOI:** 10.3390/cancers14225666

**Published:** 2022-11-18

**Authors:** Charlotte Summerfield, Lesley Smith, Oliver Todd, Cristina Renzi, Georgios Lyratzopoulos, Richard D. Neal, Daniel Jones

**Affiliations:** 1Academic Unit of Primary Care, University of Leeds, Leeds LS2 9JT, UK; 2Clinical and Population Science Department, School of Medicine, University of Leeds, Leeds LS2 9JT, UK; 3Academic Unit for Ageing and Stroke Research, University of Leeds, Leeds LS2 9JT, UK; 4Bradford Institute for Health Research, Bradford Teaching Hospitals NHS Trust, Bradford BD9 6RJ, UK; 5Epidemiology of Cancer Healthcare Outcomes (ECHO) Research Group, Research Department of Behavioural Science and Health, University College London, London WC1E 6BT, UK; 6Faculty of Medicine, University Vita-Salute San Raffaele, Milan, Via Olgettina 58, 20132 Milan, Italy; 7Department of Health and Community Sciences, Faculty of Health and Life Sciences, University of Exeter, Exeter EX4 4PY, UK

**Keywords:** cancer, diagnosis, primary care, frailty, older age

## Abstract

**Simple Summary:**

Over 60% of cancer diagnoses in the United Kingdom (UK) are in patients aged 65 and over. It can be more complicated to identify and diagnose cancer in older people due to frailty. Frailty means that people are more vulnerable and often take longer to recover from health problems. Delays in diagnosis and treatment of cancer can lead to poorer patient outcomes. Using a database of patients from the Bradford district, we identified a group of patients who presented to their GP with signs suggestive of cancer, and who went on to be diagnosed with cancer, and calculated their degree of frailty using the electronic frailty index (eFI). We found that 1 in 5 of these patients were identified as frail and that frailty was associated with a significantly longer time to cancer diagnosis. We recommend further research takes place to explore reasons behind this delay in diagnosis.

**Abstract:**

Over 60% of cancer diagnoses in the UK are in patients aged 65 and over. Cancer diagnosis and treatment in older adults is complicated by the presence of frailty, which is associated with lower survival rates and poorer quality of life. This population-based cohort study used a longitudinal database to calculate the time between presentation to primary care with a symptom suspicious of cancer and a confirmed cancer diagnosis for 7460 patients in the Bradford District. Individual frailty scores were calculated using the electronic frailty index (eFI) and categorised by severity. The median time from symptomatic presentation to cancer diagnosis for all patients was 48 days (IQR 21–142). 23% of the cohort had some degree of frailty. After adjustment for potential confounders, mild frailty added 7 days (95% CI 3–11), moderate frailty 23 days (95% CI 4–42) and severe frailty 11 days (95% CI −27–48) to the median time to diagnosis compared to not frail patients. Our findings support use of the eFI in primary care to identify and address patient, healthcare and system factors that may contribute to diagnostic delay. We recommend further research to explore patient and clinician factors when investigating cancer in frail patients.

## 1. Introduction

Over 60% of cancers in the United Kingdom (UK) are diagnosed in patients aged 65 and over [1]. Globally, the incidence of cancer in older adults (aged 65 years and above) is predicted to double by 2035 [2]. Cancer recognition, referral, diagnosis and treatment in older adults is complicated by the presence of frailty [3].

Across all ages, early cancer diagnosis is associated with higher survival rates, increased patient satisfaction and greater treatment options [4]. The National Health Service (NHS) has three national cancer screening programmes, for cervical, breast and colorectal cancer, with upper age limits of 64, 70 and 74 years, respectively. After these ages, the service requires patients to opt-in, predominantly due to a lack of evidence for screening in patients with reduced life expectancy, and concerns regarding potential negative impacts of screening, including overdiagnosis [2]. A population-based study in the United States, found older adults had lower screening rates across all cancers [5].

Symptomatic presentation to primary care is the main mode of presentation of cancer patients in the UK [6]. Since the initial publication in 2005, National Institute for Health and Care Excellence (NICE) guidelines have provided evidence to support the identification of suspected cancer [7]. Informally known as ‘the two week wait pathway’, NICE recommends patients with signs and symptoms of suspected cancer (red flags) are seen in secondary care within two weeks of presentation. National guidelines include lower age reference points for some symptoms but have no upper age limit for referral [8]. In a review of routes to cancer diagnosis, Elliss-Brookes et al. identified that adults aged 60 years and above were more likely to be diagnosed via an emergency presentation than younger patients. The proportion of cancers diagnosed from two week wait and GP referrals reduced with increasing age. One year survival for all cancers was lower in patients diagnosed via an emergency presentation [9].

Frailty is defined as a state of increased vulnerability secondary to age-associated physiological decline in reserves, ultimately compromising the ability to cope with acute stressors [10]. The electronic frailty index (eFI) is a measure of frailty which has been developed and externally validated in routine electronic health record data and is now available in all UK suppliers of GP electronic health record systems [11,12]. The eFI is based on the cumulative deficit model of frailty which characterises frailty as an accumulation of a range of health and social care related variables—termed ‘deficits’ [13]. Pre-defined ranges then categorise patients’ total scores into ‘fit/not frail’, ‘mild frailty’, ‘moderate frailty’ or ‘severe frailty’ [11].

Research suggests that frail patients with cancer have poorer outcomes. A systematic review of the prevalence of frailty in cancer patients found 42% of patients with a cancer diagnosis were classified as frail. The review found that patients with frailty had increased morbidity and mortality and were less likely to tolerate chemotherapy and radiotherapy due to increased side effects such as toxicity [14]. Frail patients undergoing elective cancer surgery were found to have an increased risk of post-operative complications such as readmission and further operations [15]. A systematic review exploring the effect of time to cancer diagnosis on patient outcomes identified improved survival and quality of life in patients with a shorter time to diagnosis for several cancer types, including breast, colorectal and prostate [16]. It is possible that the poorer cancer outcomes seen in frail patients could, in part, be due to prolonged time to diagnosis, but to date no research has considered this. We aim to review the association between old age and frailty on the time to cancer diagnosis.

## 2. Materials and Methods

### 2.1. Study Design

This population-based cohort study used the Connected Bradford dataset. The dataset is a longitudinal database containing near real-time anonymised medical records of approximately 868,000 residents in the Bradford district. It includes linked primary and secondary care data from 86 general practices and five NHS Trusts [17]. This study has been reported according to the RECORD statement [18].

### 2.2. Setting

Cohort participants include all inhabitants of Bradford, England’s fifth largest metropolitan district with urban areas among the most deprived in the country [19]. Bradford has high levels of ethnic diversity, with the largest proportion of people of Pakistani ethnic origin in England [20].

### 2.3. Participants

We studied adults presenting to primary care services with a clinically recognised sign or symptom that could be due to cancer and a subsequent diagnosis of cancer pertaining to that specific symptom (for example ‘breast lump’ leading to ‘breast cancer’ or ‘haemoptysis’ leading to ‘lung cancer’), over a 13-year period from 1 January 2009 to 25 October 2021. These signs and symptoms are listed by cancer type in NICE [NG12] recognition and referral of suspected cancer guidelines [8]. Data were collected from 2009 onwards due to the availability of primary care referral data [21].

Patients were included in the analysis if they met the following inclusion criteria

aged 18 years and above,registered with a general practice contributing data to the Connected Bradford databasepresentation to primary care with a sign or symptom that could be due to cancer [8], known informally as a ‘red flag symptom’ [22], and subsequent diagnosis of cancer.

Patients who were diagnosed via screening methods were excluded (defined as having a screening Read/SNOMED CT code prior to presentation). Patients with a time from first presentation with a red flag symptom to cancer diagnosis of more than one year were excluded, in line with other research in this field [22].

### 2.4. Data Sources

The Connected Bradford database includes information about medical diagnoses, presentations to primary care, clinical investigations, and patient demographics. Extensive professional engagement with senior stakeholders and public engagement across Bradford has resulted in support to facilitate medical research to improve healthcare services across the region. Connected Bradford has an ethical and confidentiality advisory group approval to be used for research and utilises a steering group to review data usage requests [23].

### 2.5. Variables

Externally validated Read and SNOMED CT codes were used to identify the variables within the Connected Bradford database. These codes form part of a standardised, multilingual vocabulary of clinical terminology used by health professionals to record clinical information [24]. Validated code lists were provided by separate research groups investigating early cancer diagnosis and frailty [11,25,26,27].

### 2.6. Time to Diagnosis

Externally validated Read and SNOMED CT codes were used to identify red flag symptoms and corresponding cancer types within the database. The index consultation was defined as the first time the patient presented to primary care with a red flag symptom (e.g., a patient presenting for the first time with rectal bleeding). The date of cancer diagnosis was defined as the date the cancer diagnosis was recorded in the GP electronic health record (e.g., a Read/SNOMED CT code of ‘metastatic colon cancer’). The time to diagnosis was defined as the time between the index consultation and date of diagnosis. This method has been previously used in primary care time to diagnosis research [16]. Patients with multiple primary malignancies were included in the dataset.

### 2.7. Frailty

Externally validated SNOMED CT and Read codes were also used to identify ‘deficits’ contributing to the eFI score from the GP electronic health record. This included diagnoses of chronic conditions such as heart failure and chronic kidney disease as well as social factors, cognitive impairment, and mobility problems. Individual eFI scores were calculated by combining the cumulative number of deficits at the time of presentation and dividing the total by 36. The scores were then categorised using the corresponding eFI range; not frail (0–0.12), mild frailty (>0.12–0.24), moderate frailty (>0.24–0.36), severe frailty (>0.36) [11].

### 2.8. Demographics

Patient demographics included sex, age at time of presentation (categorised as “18–64” years, “65–74”, “75–84” and “85–104”) cancer type (listed in Table 1), ethnicity (ethnic groups as defined by the 2021 UK census) and Index of Multiple Deprivation (IMD) decile [28]. These variables were chosen in accordance with previous work in this field [29]. The IMD is a measure of relative deprivation used by the Office of National Statistics to categorise areas of England. The 1st decile corresponds to the most deprived areas of England and the 10th decile relates to the least deprived areas [30].

### 2.9. Statistical Methods

Descriptive statistics were used to compare the distributions of time to diagnosis by age and level of frailty. The distribution of diagnostic interval was checked, whether normal or non-normal, to determine whether respective parametric or non-parametric summary measures should be applied. As the distribution was positively skewed, quantile regression was used to assess the association between age and frailty on time to cancer diagnosis, comparing the median, 75th and 90th centiles. Quantile regression was used to examine multiple percentiles of the distribution of time to diagnosis by age and frailty, rather than linear regression which describes differences in the mean time to diagnosis between groups only. Furthermore, there are no censored observations which would have made the use of time to event models, such as the Cox proportional hazard model, appropriate. Quantile regression does not rely upon the proportional hazards assumption and links the whole distribution to the exposures of interest. Its use in time interval data, which are always skewed, is well established [31].We performed unadjusted quantile regression and then adjusted for confounding variables—sex, ethnicity, IMD decile and cancer type. Missing data was identified for the variables ethnicity and IMD, this was addressed by multiple imputation using chained equations to produce twenty imputed datasets. Pooled estimates and 95% confidence intervals were generated using Rubin’s rule [32]. A sensitivity analysis comparing the main imputed analysis with complete case analysis was conducted. Results presented in the paper are from the imputed models. Interactions between frailty and age and frailty and cancer type were tested for statistical significance. Statistical analysis was performed using Stata/MP 17 software [33].

## 3. Results

Our study identified 7460 eligible patients with a cancer diagnosis following presentation to primary care with a symptom suggestive of cancer between the dates 1 January 2009 and 25 October 2021. Cohort demographics are listed in Table 1 (below). 48% (*n* = 3554) of the population were female. The median age was 71 years (range 18–101). 23% of the cohort (*n* = 1741) had some degree of frailty; 19% (*n* = 1401) were mildly frail, 4% (*n* = 288) moderately frail and 1% (*n* = 52) severely frail. 21% of patients fell into the 1st IMD decile, representing the 10% most deprived areas in England [30].

The median time to cancer diagnosis was 48 days (interquartile range (IQR) 21–142). We identified 18 different cancer types in the population. The most common were lung, breast, prostate and colorectal cancer, accounting for 70% of our cohort. Figure 1—below—shows marked differences in the median time to diagnosis rate between these cancers, with breast having the shortest median time to diagnosis at 21 days. Lung cancer had the longest median time to diagnosis, at 126 days. The median time to diagnosis for colorectal and prostate cancers was 38 and 75 days, respectively.

Increasing frailty was associated with substantially longer times to cancer diagnosis, with moderate and severely frail patients taking the longest time to reach a diagnosis. Patients within the not frail group had a median time to diagnosis of 43 days (IQR 21–123). The median time to diagnosis was 64 days (IQR 24–193) for patients with mild frailty and 92 days (IQR 28–232) for patients with moderate frailty. The median time to diagnosis for patients with severe frailty was 87 days (IQR 24–211). This is shown in Figure 2 (below).

Quantile regression compared the time to diagnosis by frailty score and age. These results were then adjusted for potential confounders including sex, ethnicity, cancer type and IMD. The results are shown in Table 2 (below). Mild and moderate frailty were significantly associated with a longer time to diagnosis at the median, 75th and 90th centile in both the unadjusted and adjusted results. Mild frailty added 21 days (95% CI 15–27) and moderate frailty added 49 days (95% CI 23–75) to the median time to diagnosis compared to not frail patients. Severe frailty added 44 days (95% CI −13–101) but this was not statistically significant. After adjustment for potential confounders, mild frailty added 7 days (95% CI 2–11) and moderate frailty added 23 days (95% CI 4–42) to the median time to diagnosis compared to not frail patients. After adjustment, severe frailty added 11 days (95% CI −27–48) to the median time to diagnosis however this was not statistically significant, likely as a result of small numbers in this group (*n* = 52). Increasing age was not associated with time to diagnosis after adjustment for potential confounding variables.

Testing for two-way interactions identified a significant interaction between frailty and age (*p* = 0.010) and frailty and cancer type (*p* = 0.000). Table 3 (below) shows the median time to diagnosis by level of frailty, stratified by age group. Moderate frailty had the greatest effect on time to diagnosis for the 18–64 and 65–74 years age groups, delaying median time to diagnosis by 171 days (95% CI 75–266) and 97 days (95% CI 42–152), respectively compared with the not frail group.

Table 4 (below) shows the median time to cancer diagnosis by level of frailty, stratified by cancer type for the four most common cancers in our cohort—breast, colorectal, lung and prostate. For lung cancer, median time to diagnosis was increased by 50 days (95% CI 21–78) for patients with mild frailty, 90 days (95% CI 46–134) for patients with moderate frailty and 115 days (95% CI 23–207) for patients with severe frailty. In contrast, the median times to diagnosis for patients with breast cancer were not significantly affected by any level of frailty.

## 4. Discussion

This is the first large primary care health records study to consider the effect of frailty on time to diagnosis of cancer. Analysis of data from 7460 patients in the Connected Bradford dataset has shown that frailty is associated with a significantly longer time to diagnosis of cancer. The median time to cancer diagnosis was 43 days for not frail patients, 64 days for patients with mild frailty and 92 days for patients with moderate frailty. This was statistically significant for the median, 75th and 90th quartiles when confounders were adjusted for. Whilst it is likely that this trend continues for patients with severe frailty, the small numbers of severely frail patients in our cohort meant this could not be confirmed.

Patients with mild and moderate frailty had a median adjusted increase in time to diagnosis of 7 days and 23 days, respectively, with patients in the 75th and 90th centiles having much longer delays than patients who were not frail. These increases in time to diagnosis are likely to impact negatively on cancer outcomes for these patients, reducing the chance of curative treatment, and increasing morbidity and mortality [16].

The population shift in time to diagnosis for frail patients required to achieve the increase in time to diagnosis shown in this study is significant. Studies have shown that even small delays in cancer diagnosis and treatment can increase patient mortality. Sud et al. modelled the effect of delays in cancer treatment due to delayed primary care presentation during the COVID-19 pandemic. Sud et al. reported a significant number of additional lives, and life years, lost as a result of even minor delays in diagnosis and treatment [34]. Hanna et al. considered delays in commencing cancer treatment and report that even a 28 day delay in commencing treatment (including surgical, systemic and radiotherapy) was associated with increased mortality [35]. These studies highlight that any delay in cancer treatment has a real risk of a patient’s tumour progressing from curable to non-curable, and with that a significant reduction in life expectancy. The delay in diagnosis for frail patients identified in our study warrants further investigation and action to address this disparity.

Our unadjusted results identify a statistically significant effect of frailty and age as individual variables on time to diagnosis. However, the association between age and time to diagnosis is attenuated after adjustment for frailty. We explored the significance of this with a two-way interaction analysis between frailty and age. Our subsequent age-stratified quantile regression showing the median time to diagnosis by level of frailty, found that moderate frailty disproportionately affected younger groups. This is likely to be a result of very small numbers in these sub-groups. The eFI has not been externally validated in younger populations and as such further research is recommended to explore potential reasons for this delay in diagnosis.

We identified a significant two-way interaction between frailty and cancer type. To explore this further we performed quantile regression stratified by cancer type, showing the median time to diagnosis by level of frailty for the four most common cancers in our cohort. For lung cancer, the median time to diagnosis was significantly longer for all levels of frailty compared with non-frail patients. Lung cancer can present with a variety of non-specific symptoms such as cough and shortness of breath which may initially be explained by infection, heart failure or drug side effects or present with insidious symptoms such as fatigue and weight loss all of which may be more common in frail patients and thus may delay diagnosis. The median times to diagnosis for patients with breast cancer were not significantly affected by any level of frailty. This could possibly be explained by the breast cancer symptoms such as breast lump, being largely unaffected by frailty.

### 4.1. Why Does Increasing Frailty Delay Time to Diagnosis of Cancer?

Patient, physician and system factors could all contribute to delays in diagnosis of cancer in patients with frailty. Firstly, it is possible that frail patients as a result of shared decision making or patient preference opt for a period of watchful waiting or symptom management prior to investigation or referral, which may delay diagnosis.

Existing co-morbidities or normal signs and symptoms of ageing may mask potential red flag symptoms of cancer, impairing both the patient’s and clinician’s recognition of these, thereby increasing time to referral [36]. For example, the eFI includes heart failure as a contributing factor to frailty [11]. A frail patient or health care professional may attribute new shortness of breath to existing heart disease rather than recognise this as a sign of cancer. Co-morbidities could also delay or change investigations, for example patients with heart failure may have difficulty lying flat for scans and patients with chronic kidney disease may not tolerate the contrast required for diagnostic imaging.

Finally, there are likely to be multiple system factors which could delay investigation and diagnosis in frail patients. Patients with frailty may need help with activities of daily living and transport. As such, attending primary care and hospital appointments could be challenging and may be delayed. Frail patients may have carer responsibilities for spouses with co-morbidities and as a result be unable to attend appointments. Letters sent to frail patients with appointment or clinical information may be more difficult to understand due to cognitive or visual impairments.

### 4.2. Comparison with Existing Literature

Our study identified 7460 patients with a cancer diagnosis having presented to primary care with a symptom suggestive of cancer. The cancer incidence in the dataset seems concordant with national data. In 2020, Cancer Research UK (CRUK) reported 1600 cancers cases per year within the Bradford District clinical commissioning group (CCG) [37]. Applying this to the thirteen years studied, we would estimate a total of 20,800 cancer cases in the region. Elliss-Brookes et al. established that 47% of cancer cases are identified from primary care referrals, giving an estimated total of 9776 cancer cases for our cohort over the studied time period [9].

Just under one third of our cohort of older cancer patients had some degree of frailty. This is in comparison with findings from a systematic review which reported the prevalence of frailty as 42% within older cancer patients [14]. This reduced number of frail patient’s is likely due to differences in the Bradford population, which are discussed below.

The incidence of cancer type in our cohort reflects national distributions with breast, colorectal, lung and prostate cancer accounting for the majority of cases [38]. Nationally, these cancers account for 53% of new cases compared to 70% in our population [39]. This is likely to be explained by the higher incidence of lung cancer in our cohort than national rates [40]. A probable explanation is that a larger proportion of adults in Bradford smoke cigarettes, compared with national averages [37].

Our study used the date of cancer diagnosis as recorded within the GP electronic health record. Cancer diagnosis validity has been compared between primary care and cancer registry databases in a population cohort study and concluded that recording of cancer diagnosis and mortality in primary care electronic records is generally consistent with cancer registry databases in England [41].

Handforth et al. undertook a systematic review of patients who had been diagnosed with cancer. This review found that frail cancer patients experienced significantly reduced treatment options and increased morbidity and mortality compared to patients with cancer who were not frail [14]. The results of our study may go some way to explaining these earlier findings. These findings are also supported by a systematic review by Dai et al. who showed frailty increases mortality and toxicity to therapeutic treatments in patients with lung cancer [42]. A report by CRUK suggested there were inconsistencies in treatments available to older people with cancer. The report highlighted older patients with cancer were less likely to undergo major surgical resections, receive radiotherapy or be offered chemotherapy with curative intent [43]. If patients with frailty have a longer time to diagnosis, cancer is more likely to be at an advanced stage at diagnosis resulting in fewer treatment options and increased morbidity and mortality.

### 4.3. Strengths and Limitations

This study is one of the first to use the Connected Bradford dataset [17]. We have shown this dataset can be used successfully in the study of cancer diagnosis. This is also the first study within the early cancer diagnosis literature to consider the role of frailty and to use the electronic frailty index to assess this link. Within the early cancer diagnosis field this is a novel study which provides opportunities for further research [11,25,26].

The main limitations of the study surround the sample population. Bradford is the youngest city in the UK, with 26% of the population aged below 18 years [44]. High levels of deprivation and ethnic diversity are not necessarily reflective of the UK as a whole and therefore may affect the generalisability of results. We addressed these limitations by adjusting for ethnicity and deprivation in our statistical model. Our main exposure of interest, frailty, is most prevalent amongst older people, which may lead to underrepresentation within our dataset.

The eFI was validated in older adults and its use in younger populations within this study is exploratory. There is evidence that frailty exists in younger patient cohorts [45], and a rapid review suggests frailty measures have predictive value for mortality and future hospitalisations in younger populations [46].

Our dataset had a proportion of missing data, both for ethnicity and deprivation, however, this was overcome using multiple imputation methods. The proportion of missing data for patient ethnicity may relate to the length of time the patient has been registered at the practice. A study exploring completeness of ethnicity data in primary care identified poor socioeconomic data collection prior to the 2006 introduction of remuneration under the quality and outcomes framework (QOF) [47]. The reasons behind the proportion of missing deprivation data in our cohort (8%, *n* = 575) are unclear but may correspond to patients with no fixed address, temporary visitors to the region, practice concordance with socioeconomic data collection and data sharing agreements.

### 4.4. Implications for Practice, Policy and Research

This study reports that patients with mild and moderate frailty have longer times to cancer diagnosis and this at least partially explains the consistent findings in the literature that patients with frailty have poorer cancer outcomes [3,14].

Objective measures of frailty such as the eFI, allow patients’ frailty scores to be readily and easily identifiable in primary care. In the UK, a patients’ eFI score is automatically calculated by the electronic patient health record. Evidence suggests objective assessments of frailty are being used in primary care to identify vulnerable patients, reduce hospital admissions, and improve integrated care [48,49,50]. It is possible that a patient’s eFI score could be used proactively during the diagnostic period. A high frailty score could prompt clinicians to consider and address possible barriers to investigation or referral for cancer symptoms. Issues with transport could be addressed earlier, and difficulties with, or contraindications to investigations could be considered and addressed, such as attempting to optimise kidney function prior to scans or requesting specific appointment times to improve accessibility for patients.

A large systematic review identified interventions to delay and reverse frailty. The review highlighted interventions including physical activity, health education, nutrition supplementation, home visits, hormone supplementation, and counselling and found that 71% of studies improved frailty status [51]. Much like ‘prehabilitation’ is now widely used in pre-operative assessments and prior to surgery, it is possible that by identifying frailty at a patient’s presentation with red flag symptoms, a period of ‘prehabilitation’ within primary care could improve time to diagnosis and overall outcomes [52]. The clinician or multidisciplinary team may be able to address factors such as polypharmacy, provide support for visual and hearing impairments or assist with mobility problems.

Community geriatricians and a multidisciplinary team approach could also play a role in the management of frail patients with cancer symptoms. Specialist assessment at the time of presentation could identify and help to overcome potential barriers to investigation and referral and recommend the most appropriate investigations or referral pathways for different degrees of frailty. Occupational and physiotherapy could support ‘prehabilitation’, identify and help to optimise frail patients’ independence with activities of daily living and recommend community services to support frail patients during their diagnostic pathway.

Utilising data from similar databases across the country would create a larger sample size with more widespread representation, improving reliability of results. Our findings show considerable variation from national guidelines and suggest further research is needed into decisions to investigate cancer in frail adults. Qualitative studies with frail patients and clinicians—such as interviews or discrete choice experiments—could help to explain the effect of frailty on patient factors and decisions to accept or decline investigations or referrals. Qualitative studies could also explore the system factors which affect these patients and how clinicians take frailty into account when advocating investigations and/or referrals.

## 5. Conclusions

To conclude, our study has highlighted a statistically significant delay in time to cancer diagnosis for patients with mild and moderately frailty. Our findings support use of the eFI in primary care to identify and address patient, healthcare and system factors that may contribute to diagnostic delay. We recommend further research to explore patient and clinician factors when investigating cancer in frail patients.

## Figures and Tables

**Figure 1 cancers-14-05666-f001:**
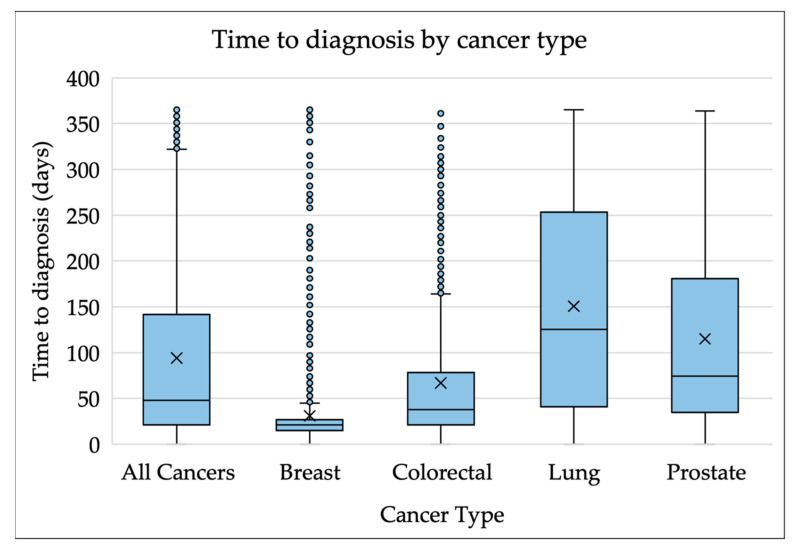
Box plots of time to diagnosis by cancer type. X represents the mean value.

**Figure 2 cancers-14-05666-f002:**
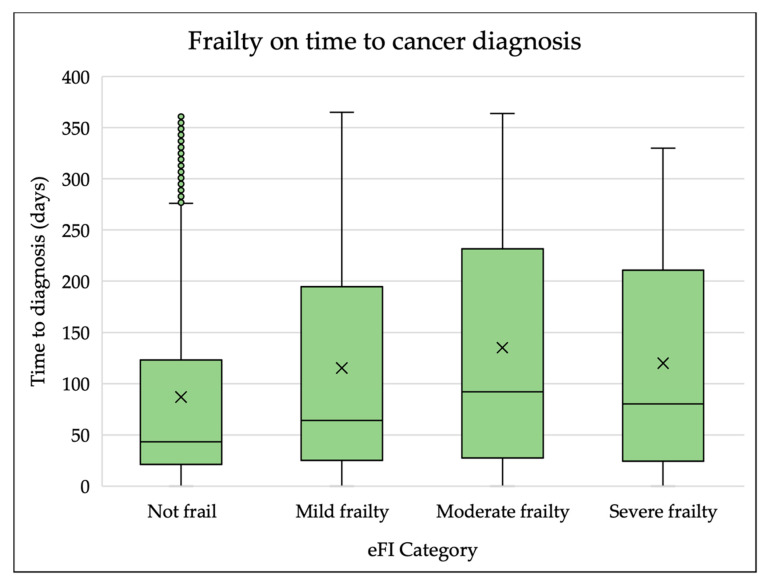
Box plots of time to cancer diagnosis by level of frailty. X represents the mean value.

**Table 1 cancers-14-05666-t001:** Cohort demographics, categorised by level of frailty.

	Total (%)		Not Frail	Mild Frailty	Moderate Frailty	Severe Frailty
	*n* = 7460		5719	1401	288	52
**Sex**						
Female	3554	47.64%	2728	648	147	31
**Age Category (years)**						
18–64	2516	33.73%	2377	132	<10 **	<10
65–74	2077	27.84%	1705	320	47	<10
75–84	1973	26.45%	1249	585	119	20
85–104	894	11.98%	388	364	117	25
**Ethnicity**						
Asian or Asian British	547	7.33%	410	111	22	<10
Black, Black British,	47	0.63%	37	<10	<10	<10
African or Caribbean						
Mixed or multiple ethnic groups	2542	34.08%	1936	486	99	21
Other ethnic group	40	0.54%	34	<10	<10	<10
White	3247	43.53%	2471	621	135	20
Missing	1037	13.90%	831	173	28	<10
**Index of Multiple Deprivation (IMD) Decile**						
1 and 2	2351	31.51%	1749	466	113	23
3 and 4	1474	19.76%	1124	280	59	11
5 and 6	1180	15.82%	916	220	39	<10
7 and 8	1004	13.46%	806	165	29	<10
9 and 10	876	11.74%	682	165	25	<10
Missing	575	7.71%	442	105	23	<10
**Cognitive impairment**						
No dementia diagnosis	7117	95.40%	5563	1278	240	36
Dementia diagnosis	343	4.60%	156	123	48	16
**Cancer Type**						
Lung	1502	20.13%	1037	368	83	14
Breast	1408	18.87%	1151	201	47	<10
Prostate	1284	17.21%	988	252	40	<10
Colorectal	1049	14.06%	842	170	30	<10
Bladder/Urinary Tract	819	10.98%	607	174	33	<10
Pancreatic	233	3.12%	161	52	15	<10
Ovarian	224	3.00%	201	19	<10	<10
Oesophageal	177	2.37%	128	40	<10	<10
Brain	175	2.35%	143	24	<10	<10
Melanoma	129	1.73%	108	17	<10	<10
Renal	116	1.55%	75	32	<10	<10
Lymphoma	98	1.31%	81	12	<10	<10
Gastric	75	1.01%	55	16	<10	<10
Endometrial	68	0.91%	60	<10	<10	<10
Cervical	36	0.48%	32	<10	<10	<10
Head and Neck	31	0.42%	23	<10	<10	<10
Myeloma	10	0.13%	<10	<10	<10	<10
Other *	26	0.35%	<25	<10	<10	<10

* Other includes testicular, plasmacytoma, leiomyosarcoma, cystic mucinous and serous tumours. ** Totals less than 10 censored to ensure no patient identifiable data present.

**Table 2 cancers-14-05666-t002:** Time to diagnosis (days) by frailty score and age category.

		Unadjusted Difference			Adjusted Difference *		
		Median	75th Centile	90th Centile	Median	75th Centile	90th Centile
		(95th CI)			(95th CI)		
***n* = 7460 (%)** **Electronic frailty index (eFI)**							
Not frail	5719 (76.66)	Ref.	Ref.	Ref.	Ref.	Ref.	Ref.
Mild frailty	1401 (18.78)	**21 (15–27)** ** *p* = 0.000	**70 (50–90)** *p* = 0.000	**61 (48–74)** *p* = 0.000	**7 (2–11)** *p* = 0.003	**21 (9–34)** *p* = 0.001	**24 (11–37)** *p* = 0.000
Moderate frailty	288 (3.86)	**49 (23–75)** *p* = 0.000	**107 (76–138)** *p* = 0.000	**74 (55–93)** *p* = 0.000	**23 (4–42)** *p* = 0.019	**51 (25–76)** *p* = 0.000	**42 (25–59)** *p* = 0.000
Severe frailty	52 (0.70)	44 (−13–101) *p* = 0.130	**75 (13–137)** *p* = 0.018	**58 (2–114)** *p* = 0.043	11 (−27–48) *p* = 0.581	45 (−9–100) *p* = 0.103	8 (−23–38) *p* = 0.621
Constant	-	**43**	**123**	**250**	**64**	**150**	**280**
**Age (years)**							
18–64	2516 (33.73)	Ref.	Ref.	Ref.	Ref.	Ref.	Ref.
65–74	2077 (27.84)	**18 (14–22)** *p* = 0.000	**58 (43–73)** *p* = 0.000	**45 (24–66)** *p* = 0.000	−1 (−3–2) *p* = 0.572	0 (−5–5) *p* = 0.912	1 (−11–13) *p* = 0.853
75–84	1973 (26.45)	**24 (19–29)** *p* = 0.000	**69 (52–86)** *p* = 0.000	**52 (33–71)** *p* = 0.000	1 (−1–4) *p* = 0.345	1 (−5–6) *p* = 0.831	−3 (−15–8) *p* = 0.563
85–101	894 (11.98)	**19 (12–26)** *p* = 0.000	**73 (45–101)** *p* = 0.000	**57 (37–77)** *p* = 0.000	−1 (−5–3) *p* = 0.593	2 (−7–11) *p* = 0.647	−9 (−23–6) *p* = 0.247
Constant	-	**35**	**99**	**235**	**64**	**150**	**280**

* adjusted for frailty, age, sex, ethnicity, cognitive impairment, cancer type and IMD decile. ** estimates in bold are statistically significant with *p* < 0.05.

**Table 3 cancers-14-05666-t003:** Median time to diagnosis (days) by level of frailty, stratified by age.

	Age Category (Years)			
	**18–64**	65–74	75–84	85–104
	**(95th CI)**			
**eFI**				
Not frail	Ref.	Ref.	Ref.	Ref.
Mild frailty	1 (−7–9) *p* = 0.803	8 (−3–18) *p* = 0.214	7 (−1–14) *p* = 0.064	8 (0–15) *p* = 0.065
Moderate frailty	**171 (75–266)** * *p* = 0.001	**97 (42–152)** *p* = 0.001	16 (−16–48) *p* = 0.317	10 (−2–23) *p* = 0.170
Severe frailty	20 (−19–59) *p* = 0.303	5 (−97–107) *p* = 0.909	17 (−52–87) *p* = 0.637	9 (−51–68) *p* = 0.800
Constant	**51**	**69**	**66**	**70**

* estimates in bold are statistically significant with *p* < 0.05.

**Table 4 cancers-14-05666-t004:** Median time to diagnosis (days) by level of frailty, stratified by cancer type.

	Cancer Type			
	Breast	Colorectal	Lung	Prostate
	(95th CI)			
**eFI**				
Not frail	Ref.	Ref.	Ref.	Ref.
Mild frailty	0 (−1–1) *p* = 0.805	**11 (0.36–22)** * ***p* = 0.043**	**50 (21–78)** ***p* = 0.001**	**32 (3–60)** ***p* = 0.028**
Moderate frailty	1 (−3–5) *p* = 0.611	19 (−18–56) *p* = 0.324	**90 (46–134)** ***p* = 0.000**	52 (−15–119) *p* = 0.130
Severe frailty	−2 (−9–5) *p* = 0.577	8 (−65–82) *p* = 0.827	**115 (23–207)** ***p* = 0.015**	−19 (−155–117) *p* = 0.786
Constant	**20**	**35**	**79**	**73**

* estimates in bold are statistically significant with *p* < 0.05.

## Data Availability

Coding available for study on request to the authors, all statistical data has been published.

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
