# Peer review of "The Effect of Older Age and Frailty on the Time to Diagnosis of Cancer: A Connected Bradford Electronic Health Records Study"

_cancers, 2022, doi:10.3390/cancers14225666_

Round 1

Reviewer 1 Report

CANCERS-2021947 presents results for cancer detection in persons with frailty. I hope the authors consider my minor feedback.

·         Line 82: Recommend avoiding “explore” given that this suspects the work was not hypothesis-driven.

·         Participants: Would recommend restricting the sample to 65+ years given the Introduction focused on older adults and frailty is mostly an age-related condition.

·         Were any other covariates considered for the analyses? Those included seem sparse given the sample and source.

·         Line 270: Remove results from a discussion here and elsewhere.

·         Discussion: Can the authors comment, perhaps in this section, why time to event analyses (e.g., survival analysis) were not conducted?  

·         Make any changes to the abstract that align with those from the text.

Reviewer 2 Report

the Authors present an exceptionally well-conducted database study on the association between older age/frailty with delay in cancer diagnosis. thanks for bringing this information to the light.

I was impressed by the soundness of the findings and the quality of presentation. My only comment is: can the Authors expand on the actual relevance of a some day-delay in cancer diagnosis on ultimate outcomes? Indeed, to support their speculations (on which I agree, on a personal note), they present the results of one study by Sud et al (ref [33]), which specifically refers to the earlier phases of the COVID pandemic, with all the inherent difficulties of that period. Maybe some more comments on this can be a welcome addition to the manuscript.
